# PEER: A Collaborative Language Model

**Timo Schick**[1]    **Jane Dwivedi-Yu**[1]    **Zhengbao Jiang**[1,2]    **Fabio Petroni**[1]
**Patrick Lewis**[1]    **Gautier Izacard**[1,3]    **Qingfei You**[1]    **Christoforos Nalmpantis**[1]
**Edouard Grave**[1]    **Sebastian Riedel**[1,4]

[1] Meta AI Research    [2] Carnegie Mellon University
[3] Inria & ENS, PSL University    [4] University College London

## ABSTRACT

Textual content is often the output of a collaborative writing process: We start with an initial draft, ask for suggestions, and repeatedly make changes. Agnostic of this process, today's language models are trained to generate only the final result. As a consequence, they lack several abilities crucial for collaborative writing: They are unable to update existing texts, difficult to control and incapable of verbally planning or explaining their actions. To address these shortcomings, we introduce PEER, a *collaborative* language model that is trained to imitate the entire writing process itself. PEER can write drafts, add suggestions, propose edits and provide explanations for its actions. Crucially, we train multiple instances of PEER able to *infill* various parts of the writing process, enabling the use of self-training techniques for increasing the quality, amount and diversity of training data. This unlocks PEER's full potential by making it applicable in domains for which no edit histories are available and improving its ability to follow instructions, to write useful comments, and to explain its actions. We show that PEER achieves strong performance across various domains and editing tasks.

## 1 INTRODUCTION

Large neural networks show impressive text generation capabilities when pretrained with a language modeling objective (Radford et al., 2019; Raffel et al., 2020; Brown et al., 2020; Rae et al., 2021; Zhang et al., 2022; Chowdhery et al., 2022, i.a.). However, the way these models operate – producing outputs in a single pass from left to right – differs strongly from the iterative process by which humans typically write texts. This limits their utility for *collaborative* writing in various respects; for example, they are not able to retroactively modify or refine their own outputs. Beyond that, they are hard to control (Korbak et al., 2022) and verifying their outputs is challenging as they often hallucinate content (Maynez et al., 2020; Shuster et al., 2021; Nakano et al., 2021) and lack the ability to explain their intentions. All of this makes it very difficult for humans to collaborate with such models for writing coherent, factual texts.

To address these shortcomings of existing LMs, we propose PEER (**P**lan, **E**dit, **E**xplain, **R**epeat), a *collaborative* language model trained on edit histories to cover the entire writing process. As illustrated in Figure 1, PEER operates in several steps that aim to mirror the human writing process: For a given text, either a user or the model itself can *plan* an action to be applied, for example by means of a natural language instruction. This plan is then realized by an *edit*, which the model can *explain* both in form of a textual comment and by pointing to references used; this is enabled by augmenting each input text with retrieved passages containing potentially relevant background information. We *repeat* these steps until the text is in a satisfactory state that does not require any further updates. This iterative approach does not only enable the model to decompose the complex task of writing a consistent, factual text into multiple easier subtasks, it also allows humans to intervene at any time and steer the model in the right direction, either by providing it with their own plans and comments or by making edits themselves.

Similar to recent approaches for iterative editing (Faltings et al., 2021; Reid & Neubig, 2022), we use Wikipedia as our main source of edits and associated comments, which we use as proxies for plans and explanations. In contrast to this prior work, however, our goal is to obtain a collaborative model

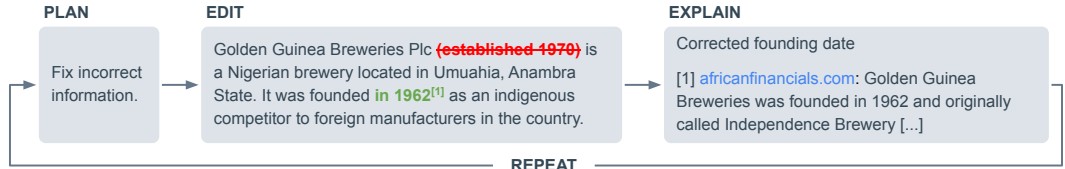

Figure 1: Illustration of the steps performed by PEER: First, either the user or the model specifies a ***plan*** describing the action they want to be performed; this action is then realized by means of an ***edit***. The model can ***explain*** the edit both in natural language and by pointing to relevant sources. We can ***repeat*** this process until the generated text requires no further updates.

that is useful *beyond* just Wikipedia: It should be capable of following human-written instructions for updating texts in any domain. To achieve this goal, we train PEER not only to perform the writing process illustrated in Figure 1 in sequential order, but also to *infill* various parts; for example, given an edited text and a set of relevant documents, we teach it to produce the original version of this text *before* it was edited. This enables us to use self-training techniques (e.g., Yarowsky, 1995; Sennrich et al., 2016; He et al., 2020a; Schick & Schütze, 2021a) for training PEER with synthetic plans, edits, explanations and documents. We show that this substantially improves PEER along several axes, including its ability to edit texts in any domain, to understand human-written instructions, and to explain its actions.

## 2   RELATED WORK

**Text Editing**   Similar to our work, Faltings et al. (2021) train an editing model to follow plans on Wikipedia data. However, they only consider single sentence edits, evaluate on Wikipedia data only and do not explore approaches for improving data quality and coverage. Reid & Neubig (2022) also train models on Wikipedia's edit history, but do not consider plans, explanations or reference documents. Several editing models are trained to solve specific tasks, such as updating information (Logan IV et al., 2021), fixing grammar errors (Napoles et al., 2017; Awasthi et al., 2019) or improving citations (Petroni et al., 2022). Various approaches teach models to iteratively improve texts in an unsupervised fashion (e.g., Shen et al., 2020; Donahue et al., 2020; Li et al., 2022) and explore more efficient ways of representing edits (Mallinson et al., 2020). Concurrent work of Dwivedi-Yu et al. (2022) proposes EDITEVAL, a benchmark for evaluating editing models.

**Instruction Tuning and Planning**   Explicitly teaching models to follow plans is related to recent work that finetunes models on human-written instructions (Wei et al., 2022a; Sanh et al., 2022; Bach et al., 2022; Ouyang et al., 2022; Wang et al., 2022). The idea of having a separate *planning* stage has also been explored for other text generation tasks inlcuding summarization (Narayan et al., 2021), data-to-text generation (Moryossef et al., 2019) and story writing (Yao et al., 2019). Our approach of writing text by iteratively performing small updates has some similarity with recent approaches like *chain-of-thought* prompting (Wei et al., 2022b; Dohan et al., 2022) and document sketching (Wu et al., 2021), that also break down a complex task into multiple smaller steps.

**Collaborative Writing**   Du et al. (2022a;b) investigate human-machine interactions for iteratively improving documents; however, they focus mostly on syntactic edits that improve the fluency, coherence or style of a document. Lee et al. (2022) investigate using GPT3 (Brown et al., 2020) as a writing assistant for creative and argumentative writing. In their setup, however, the model provides suggestions for continuations without being controllable by means of natural language instructions.

**Self-Training**   Our approach of using models to infill missing data closely resembles other self-training and bootstrapping approaches used e.g. in word sense disambiguation (Yarowsky, 1995), machine translation (Sennrich et al., 2016; Hoang et al., 2018), sequence generation (He et al., 2020a), and few-shot learning (Schick & Schütze, 2021a;b). Similar to how we use models to turn plain texts into sequences of edits, Dai et al. (2022) turn documents into dialogue sequences.

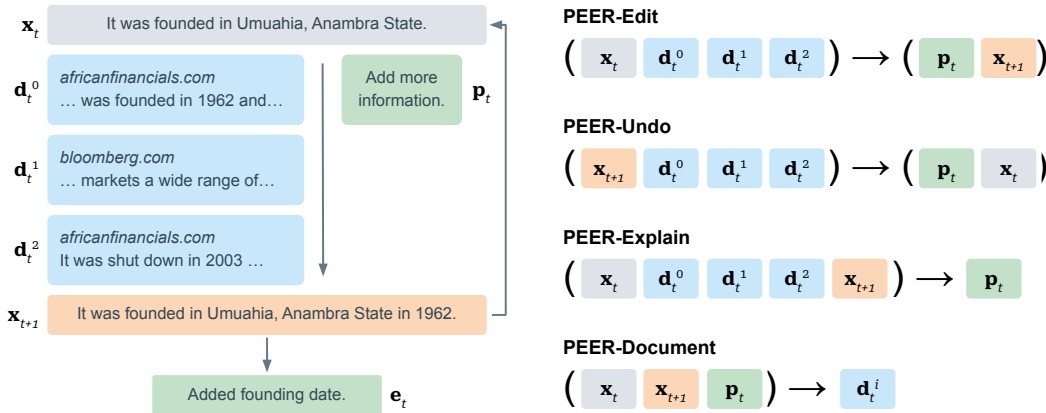

Figure 2: Schematic representation of the PEER process. **Left**: Starting from a text $\mathbf{x}_t$, we use both a plan $\mathbf{p}_t$ and a collection of documents $D_t = \{d_t^0, \ldots, d_t^k\}$ to obtain an updated version $\mathbf{x}_{t+1}$ and an explanation $\mathbf{e}_t$ of the performed edit; this process is repeated multiple times. **Right**: To generalize to domains without editing histories, overcome data scarcity and improve the model's core abilities, we train various instances of PEER that perform different infilling tasks derived from this process.

## 3  PLAN, EDIT, EXPLAIN, REPEAT

The core idea of our proposed framework is to model the editing of textual content as an *iterative process*, where we repeatedly *plan* and *realize* changes (see Figure 2, left). Each iteration within this framework edits a text sequence $\mathbf{x}_t$ to obtain an updated version $\mathbf{x}_{t+1}$. For this edit, we assume that we are given a set of documents $D_t = \{d_t^1, \ldots, d_t^k\}$ containing relevant background information.[1] Given $\mathbf{x}_t$ and $D_t$, we first formulate a *plan* $\mathbf{p}_t$ – a rough idea of how the text should be modified, verbalized as a short text sequence like "add more information", "fix grammar errors" or "use simpler language". This plan is then realized by means of an actual *edit* that transforms $\mathbf{x}_t$ into the updated state $\mathbf{x}_{t+1}$. Finally, the intention behind this edit can optionally be clarified by providing a textual *explanation* $\mathbf{e}_t$; this is especially relevant in collaborative settings where explanations can facilitate evaluating the quality and usefulness of an edit (Liu et al., 2019). Note that the explanation can be similar or even identical to the plan, the conceptual difference being that the plan is made *before* performing the edit, whereas the explanation is only formulated *after* it was performed. The entire process of formulating a plan, collecting documents, performing an edit and explaining it, can be repeated multiple times to obtain a sequence of texts $\mathbf{x}_t, \mathbf{x}_{t+1}, \mathbf{x}_{t+2}, \ldots$ until either we arrive at some $\mathbf{x}_n$ for which $\mathbf{x}_n = \mathbf{x}_{n-1}$, or we reach a manually defined halting criterion. We can also write texts from scratch by starting with an empty sequence, i.e., $\mathbf{x}_0 = \varepsilon$. In reference to its four main parts, we refer to models based on this iterative process as PEER models.

While using PEER to break the complex task of writing a coherent, consistent and factual document into many smaller subtasks has some potential benefits over standard left-to-right language modeling – such as being more interpretable and easier to control – it is challenging to find data from which this process can be learned at the scale required to train large language models. This is mainly because edit histories are difficult to obtain from web crawls, the most important data source for current language models (Brown et al., 2020; Rae et al., 2021). But even in cases where edit histories can be obtained (e.g., by collecting crawls of identical pages at different times) or synthetically generated, edits are typically not annotated with plans, documents, or explanations.

Similar to prior work on text editing (Faltings et al., 2021; Reid & Neubig, 2022), our first step in overcoming this issue is turning to Wikipedia – a single source that comes close to fulfilling all our needs: It provides a full edit history including comments on a diverse set of topics, is large in scale, and articles frequently contain citations, which can be helpful for finding relevant documents. However, relying on Wikipedia as our sole source of training data comes with various severe downsides: First, it makes trained models specific to Wikipedia in terms of how they expect textual

---

[1]This set aims to mirror the result of background research that humans may conduct before writing or editing texts. However, modeling this research itself is beyond the scope of this work, so we consider $D_t$ as given.

content to look like and what plans and edits they predict. Beyond that, comments in Wikipedia are, in many cases, not an appropriate proxy for plans or explanations. Finally, numerous paragraphs in Wikipedia do not contain any citations; while this lack of background information can often be compensated by using a retrieval system (Piktus et al., 2021; Petroni et al., 2022), even such systems may not be able to find supporting background information for many edits.

## 4 Infilling Edit Histories with PEER

We propose a simple approach to address all issues that arise from Wikipedia being our only source of commented edit histories at once: We train not just one, but multiple instances of PEER that learn to infill various parts of the editing process (Figure 2, right);[2] these models can then be used to generate synthetic data as a substitute for the missing pieces in our training corpus. In concrete terms, we train the following encoder-decoder models:

- **PEER-Edit**: Given an input text and a set of documents, this model learns to both plan and realize edits, i.e., it maps $(\mathbf{x}_t, D_t)$ to the sequence $(\mathbf{p}_t, \mathbf{x}_{t+1})$. This is done in an autoregressive fashion by factoring

$$p(\mathbf{p}_t, \mathbf{x}_{t+1} \mid \mathbf{x}_t, D_t) = \prod_{i=1}^{n} p(z_i \mid \mathbf{x}_t, D_t, z_1, \ldots, z_{i-1}) \tag{1}$$

  where $\mathbf{z} = z_1, \ldots, z_n = \mathbf{p}_t \cdot \mathbf{x}_{t+1}$ is the concatenation of $\mathbf{p}_t$ and $\mathbf{x}_{t+1}$. Thus, PEER-Edit can update texts autonomously by generating both plans and edits, but it can also be provided with human-written plans as prefixes. As PEER-Edit is our main model for actual editing, we also refer to it simply as PEER.
- **PEER-Undo**: Given a text sequence $\mathbf{x}_{t+1}$ and a collection of documents $D_t$ that may have been used to write it, this PEER instance is trained to guess and *undo* the latest edit by predicting the sequence $(\mathbf{p}_t, \mathbf{x}_t)$. This is done autoregressively analogous to PEER-Edit.
- **PEER-Explain**: This model is trained to autoregressively generate explanations $\mathbf{e}_i$ given $(\mathbf{x}_t, \mathbf{x}_{t+1}, D_t)$, i.e., an edit and a collection of relevant documents.
- **PEER-Document**: Given $(\mathbf{x}_t, \mathbf{x}_{t+1}, \mathbf{p}_t)$, this model is trained to generate a document $d \in D_t$ that provides useful background information for the edit.

We use all variants of PEER to produce synthetic data – both to generate the missing pieces for completing our training data, and to *replace* low-quality pieces in our existing data.

**Decomposing Texts**  To enable training on arbitrary text data even if it comes without edit histories, we use PEER-Undo for generating synthetic "backward" edits: Given a plain text $\mathbf{x} = \mathbf{x}_n$ and a collection of documents $D$, we iteratively apply PEER-Undo to obtain a sequence $(\mathbf{p}_{n-1}, \mathbf{x}_{n-1})$, $(\mathbf{p}_{n-2}, \mathbf{x}_{n-2}), \ldots$ until we arrive at some $\mathbf{x}_m = \varepsilon$. We can then train PEER-Edit in the opposite direction, i.e., to predict each $(\mathbf{p}_t, \mathbf{x}_{t+1})$ from $\mathbf{x}_t$ and $D$.

**Generating Plans**  We use PEER-Explain to address both the low quality of many comments in our corpus, and the fact that some edits may not have any comments. Given $\mathbf{x}_t$, $\mathbf{x}_{t+1}$ and a collection of documents $D_t$, we sample various outputs $\mathbf{e}_t^1, \ldots, \mathbf{e}_t^k$ from PEER-Explain$(\mathbf{x}_t, \mathbf{x}_{t+1}, D_t)$ that explain the edit being made and act as potential plans. We then compute the likelihood of the actual edit given each $\mathbf{e}_t^j$ and pick the one that makes this edit the most likely as its new plan:

$$\hat{\mathbf{p}}_t = \underset{j \in \{1, \ldots, k\}}{\arg\max} \, p(\mathbf{x}_{t+1} \mid \mathbf{x}_t, D_t, \mathbf{e}_t^j) \tag{2}$$

where $p(\mathbf{x}_{t+1} \mid \mathbf{x}_t, D_t, \mathbf{e}_t^j)$ is the probability that PEER-Edit assigns to $\mathbf{x}_{t+1}$ given $\mathbf{x}_t$, $D_t$ and $\mathbf{e}_t^j$.

**Generating Documents**  If we are cannot find relevant documents for an edit, we can use PEER-Document to generate synthetic ones. We only do so for *training* PEER-Edit; we never provide synthetic documents during inference. Analogous to plans, sample multiple documents from PEER-Document and pick the one that helps PEER-Edit the most in predicting the actual edit.

---

[2]In preliminary experiments, we also tried training a single model to perform all infilling tasks at once, but found this approach to perform worse.

Table 1: SARI scores on all subsets of Natural Edits. Domain-adapted (DA) variants outperform regular PEER, demonstrating the usefulness of synthetic edits generated with PEER-Undo.

|  | Wiki | News | Cook. | Garden | Law | Movies | Politics | Travel | Workpl. |
|---|---|---|---|---|---|---|---|---|---|
| Copy | 32.7 | 32.8 | 31.6 | 32.0 | 31.1 | 31.5 | 31.8 | 31.2 | 31.5 |
| PEER (no plans) | 50.7 | 41.3 | 36.3 | 35.1 | 35.8 | 35.3 | 36.5 | 34.8 | 34.7 |
| PEER | **55.5** | 49.3 | 40.2 | 37.7 | 36.4 | 39.2 | 38.7 | 38.1 | 36.7 |
| PEER (DA) | – | **51.6** | **42.9** | **44.9** | **39.0** | **42.4** | **41.3** | **40.2** | **39.2** |

## 5 EXPERIMENTS

We conduct a series of experiments to investigate whether – despite Wikipedia being our only *natural* source of comments and edits – our infilling techniques enable us to turn PEER into a general purpose editing model capable of following human-written plans and tackling a range of editing tasks in different domains. Specifically, we aim to answer the following questions:

- Can PEER follow plans and perform edits in domains for which no edit histories are available, and does self-training on decomposed texts improve this ability? (Section 5.1)

- Does the ability to follow plans based on Wikipedia comments transfer to instructions specified by humans, and can it be improved by training on synthetic plans? (Section 5.2)

- Can PEER make proper use of citations and quotes to explain generated outputs, and can PEER-Document be used to amplify this? (Section 5.3)

- How does writing texts in a single pass compare to iteratively applying PEER? (Section 5.4)

**Experimental Setup** Our main training data is based on Wikipedia's edit history. For each edit, we use citations and a retrieval system (Petroni et al., 2022) to obtain up to 3 relevant documents $D = \{d_0, d_1, d_2\}$. We replace each citation of a document $d_i$ with `[[[i]]]` or `[[[i quote=`$\mathbf{q}_i$`]]]` depending on whether a specific subsequence $\mathbf{q}_i$ of $d_i$ was quoted in the original data. We simply concatenate all inputs and outputs, separated by a special sequence. Further details are discussed in Appendix A. We initialize all instances of PEER from an existing pretrained language model with 3B parameters. Each model is trained for 20,000 steps on 64 GPUs with an effective batch size of 256, corresponding to about five million Wikipedia edits.

### 5.1 NATURAL EDITS

We first evaluate PEER's ability to follow plans and perform edits in domains for which no edit histories are available. To this end, we introduce *Natural Edits*, a collection of naturally occuring edits for different text types and domains that we obtain from three English web sources: We collect encyclopedic pages from Wikipedia, news articles from Wikinews, and questions from the *Cooking*, *Gardening*, *Law*, *Movies*, *Politics*, *Travel* and *Workplace* subforums of StackExchange. All of these sites provide edit histories with comments that often elaborate on the edit's intent and that we provide to all models as plans. We split each dataset into training and test data. However, we only provide plain texts instead of actual edits in the training sets of the Wikinews and StackExchange subsets, enabling us to test editing abilities in domains for which no edit histories are accessible. Relevant statistics for Natural Edits are shown in Table 6.

To leverage available plain texts, we use PEER-Undo as described in Section 4 and create synthetic in-domain edits on which we train domain-adapted (DA) variants of PEER. These variants are finetuned on a balanced mixture of examples from the original training distribution and synthetic edits for 1,000 steps; we do so separately for the Wikinews and StackExchange subsets of Natural Edits, resulting in two instances of domain-adapted PEER. Results shown in Table 1 illustrate that plans are extremely helpful across domains, indicating that the ability to understand plans found in Wikipedia edits directly transfers to other domains. Importantly, the domain-adapted variants of PEER outperform regular PEER for all subsets of Natural Edits. This demonstrates the effectiveness of generating synthetic edits for applying PEER in different domains.

Table 2: Downstream task results for PEER and various baselines, divided into three groups: **(a)** a copy baseline, T5-based models, and PEER **(b)** 175B parameter decoder-only models, **(c)** supervised SotA. The first numbers for each task are SARI scores; additional metrics are GLEU for JFLEG, EM for WNC and Update-R1 for FRUIT. Supervised scores from left to right are from Ge et al. (2018), Martin et al. (2020), Du et al. (2022b), Pryzant et al. (2020) and Logan IV et al. (2021), respectively. The best result for models in the first group is shown in bold, the best zero-shot performance overall is underlined. On average, PEER (SP) clearly outperforms all baselines.

| Model | Params | Without Documents | | | | With Documents | | Avg |
| | | JFLEG | ASSET | ITER | WNC | FRUIT | WAFER | |
|---|---|---|---|---|---|---|---|---|
| Copy | – | 26.7 / 40.5 | 20.7 | 30.5 | 31.9 / 0.0 | 29.8 / 0.0 | 33.6 | 28.9 |
| Tk-Instruct | 3B | 31.7 / 38.7 | 28.3 | 36.2 | 30.3 / 0.0 | 12.7 / 3.9 | 1.6 | 23.5 |
| T0 | 3B | 42.9 / 38.6 | 28.6 | 28.1 | 17.8 / 0.0 | 13.1 / 5.7 | 6.1 | 22.8 |
| T0++ | 11B | 35.9 / 43.8 | 25.8 | 36.1 | 27.0 / 0.0 | 16.1 / 3.7 | 3.9 | 24.1 |
| PEER | 3B | 54.8 / 55.1 | 29.9 | 36.5 | 56.4 / 31.9 | 39.4 / 28.3 | 35.2 | 42.0 |
| PEER (SP) | 3B | 59.0 / 57.2 | **33.2** | 37.1 | 56.6 / 32.7 | 40.3 / **33.9** | 35.5 | 43.6 |
| PEER (SP) | 11B | **59.9 / 58.6** | 32.4 | **37.8** | 58.8 / 34.7 | **40.7** / 33.5 | **35.9** | **44.3** |
| OPT | 175B | 49.2 / 49.4 | 25.8 | 31.4 | 25.1 / 0.0 | 35.6 / 27.4 | 21.1 | 31.4 |
| GPT3 | 175B | 50.6 / 51.8 | 25.0 | 30.7 | 26.0 / 0.5 | 33.6 / 25.9 | 22.9 | 31.5 |
| InstructGPT | 175B | 62.3 / 60.0 | 35.4 | 38.2 | 33.9 / 0.7 | 37.5 / 23.4 | 29.2 | 39.4 |
| Sup. SotA | – | – / 62.4 | 44.2 | 37.2 | – / 45.8 | – / 47.4 | – | – |

## 5.2 DOWNSTREAM TASKS

So far, we have evaluated PEER using plans based on naturally occurring *comments*. But to what extend is it capable of following *instructions* formulated by humans to yield well known editing functionalities, and can training on synthetic plans improve this ability? To answer these questions, we evaluate PEER on various editing tasks in a zero-shot fashion. We use the following datasets:

- **JFLEG** (Napoles et al., 2017) is a grammatical error correction dataset with single-sentence inputs written by English language learners;
- **ASSET** (Alva-Manchego et al., 2020) is a corpus for single-sentence text simplification;
- **ITERATER** (Du et al., 2022b) is an editing dataset spanning five edit intentions across three different domains;[3]
- **WNC** (Pryzant et al., 2020) is a dataset where the task is to remove or mitigate biased words to make sentences more neutral;
- **FRUIT** (Logan IV et al., 2021) contains texts from Wikipedia that need to be *updated*; for performing this update, various reference documents from Wikipedia are provided;
- **WAFER-INS** (Dwivedi-Yu et al., 2022) is based on the WAFER dataset (Petroni et al., 2022); the task is to *insert* a sentence in a Wikipedia paragraph given documents from the Sphere corpus (Piktus et al., 2021) that contain relevant background information.

In addition to PEER-Edit, we also consider a variant trained with synthetic plans; that is, we replace each original plan with one generated by PEER-Explain as described in Section 4. We refer to the PEER-Edit variant trained on these synthetic plans as PEER (SP). We compare to various baseline models: Tk-Instruct (Wang et al., 2022), T0 and T0++ (Sanh et al., 2022), three models that are initialized from the *LM Adapt* variant of T5 and finetuned on collections of manually written instructions. We also compare to OPT (Zhang et al., 2022) and GPT3 (Brown et al., 2020), two large LMs with 175B parameters, and InstructGPT (Ouyang et al., 2022), the instruction-tuned variant of GPT3.[4] We formulate a single plan **p** per task that we provide to all models (see Appendix E.2) and use greedy decoding. We do not perform any task-specific finetuning or in-context learning as we are interested in evaluating each model's suitability as a *general* editing model: In the general case of a user providing a plan, we cannot assume access to other examples using the same plan.

---

[3]We only include edits from the *non-meaning-changed* categories "fluency", "coherence" and "clarity".

[4]We use the *text-davinci-001* variant described in (Ouyang et al., 2022).

Table 3: Accuracy on NE-Cite (without/with gold positions) and R1/R2/RL scores on both NE-Quote and constrained NE-Quote. When given the correct position, PEER (SP) almost matches the performance of the supervised Side model on NE-Cite, demonstrating its strong citing abilities. Training on synthetic documents substantially improves PEER's ability to quote relevant passages.

| Model | NE-Cite | NE-Quote | NE-Quote (con.) |
|---|---|---|---|
| Random | – / 33.3 | – | 40.1 / 31.7 / 36.5 |
| Unigram | – / 34.2 | – | – |
| Side | – / **91.1** | – | – |
| Lead | – / – | – | 50.6 / 44.0 / 46.0 |
| PEER | 74.1 / 88.1 | 0.0 / 0.0 / 0.0 | 49.3 / 44.3 / 48.1 |
| PEER (SP) | 74.5 / 88.9 | 0.2 / 0.1 / 0.1 | 49.8 / 44.8 / 48.7 |
| PEER (SQ) | **74.9** / 87.9 | **13.6 / 11.9 / 12.9** | **58.1 / 54.6 / 57.3** |

Results are shown in Table 2. PEER substantially outperforms all baseline models based on *LM Adapted* T5, with the 3B model achieving an average SARI (Xu et al., 2016) score of 42.0 across all tasks, compared to 24.1 for the strongest T5-based baseline. PEER (SP) consistently outperforms regular PEER, increasing average SARI by 1.6 points. This demonstrates the usefulness of generating synthetic plans to enhance PEER's ability to follow instructions. Increasing model size to 11B improves results for most tasks. While OPT and GPT3 perform worse than PEER, InstructGPT outperforms PEER for some tasks. However, it clearly lags behind PEER when it comes to handling documents for updating text. Averaged across all tasks, it performs 4.1 points worse than PEER (SP), despite being both larger *and* finetuned on human-annotated data. Our zero-shot models lag behind the supervised SotA on average, but approach supervised performance in some cases.

## 5.3 CITATIONS AND QUOTES

Unlike our baseline models, PEER is capable of both *citing* and *quoting* from reference documents to back up the claims it generates. This is useful in terms of explainability and verifiability, as it allows users to fact-check these claims more easily; the ability to quote individual phrases – as opposed to citing an entire document – is especially helpful for long documents. To facilitate the evaluation of PEER's ability to cite and quote, we consider both tasks in isolation. To this end, we introduce two new datasets based on Natural Edits: NE-*Cite* and NE-*Quote*. For building these datasets, we create examples from Wikipedia's edit history where the only difference between $x_t$ and $x_{t+1}$ is that a new citation or quote was added, respectively. Further details are shown in Appendix E.3.

Importantly, PEER's training data contains only few quotes. This is mainly because they are used sparingly in Wikipedia. Moreover, we are unable to use the majority of edits containing quotes because they come from non-online sources or web pages that no longer exist, so we do not have access to the documents that the quotes are taken from. To overcome this issue, we use PEER-Document to write synthetic documents for all edits that add quotes and for which the actual document is missing. We finetune PEER on these examples, mixed with around 500k examples from the original distribution, for 2,000 steps; we refer to this variant trained with synthetic quotes as PEER (SQ).

For NE-Cite, we use the percentage of times where the correct document was cited *and* the citation was placed at the right position as evaluation metric. We consider three baselines: randomly picking a reference, selecting the reference that maximizes the unigram overlap with $x_t$, and using the *Side* reranker (Petroni et al., 2022), a model trained on millions of Wikipedia citations. Unlike PEER, none of these baselines is able to decide *where* to place the citation. We thus also consider a variant of NE-Cite where models are told where to place the citation; for PEER, this is achieved through constrained decoding. Scores both without and with providing the correct positions are shown in Table 3. If not provided with the correct position, PEER puts the right citation at the right place in 74.1% of cases, with PEER (SP) slightly improving performance. When given the correct position, PEER (SP) even comes close to the supervised Side model (88.9 vs 91.1), clearly outperforming the other baselines. Finetuning on synthetic quotes does not significantly alter PEER's citing ability.

Similar to citing, we also look at two variants of the quoting task: In the first variant, the model needs to add a quote without any additional information; in the second variant, it is told where to put it and which document to quote from. For this variant, we use constrained decoding (Cao et al.,

Table 4: Rouge-1/2/L and QuestEval scores for various approaches on our Wikipedia intro generation test set. Length penalty (LP) is optimized on a dev set of 100 examples. WikiLM performs better than autonomous PEER in terms of Rouge scores, but is outperformed by PEER in manual and collaborative mode; all PEER models perform better in terms of QuestEval.

| Model | LP | R1 / R2 / RL | QuestEval |
|---|---|---|---|
| Wiki-LM | 5.0 | 38.4 / 16.9 / 27.3 | 38.7 |
| PEER (autonomous) | 5.0 | 37.7 / 15.8 / 26.2 | 40.6 |
| PEER (manual) | 2.0 | 39.4 / 17.0 / 28.1 | **41.1** |
| PEER (collaborative) | 2.0 | **39.5 / 17.2 / 28.4** | 41.0 |

2021) to ensure that the generated quote is contained in the cited document. We consider two baselines: One that selects a random substring of $n$ words from the cited document, and one that selects the lead $n$ words, where $n$ is the median length of quotes in NE-Quote. Table 3 shows Rouge-1/2/L scores (Lin, 2004) computed only on quotes. As can be seen, PEER and PEER (SP) are unable to quote without constrained decoding. Training on synthetic documents improves performance, but still results in low scores. Constrained decoding improves performance, but PEER still does not outperform the lead baseline. However, PEER (SQ) achieves much stronger results in this setting, improving R1/R2/RL scores by 7.5, 10.6 and 11.3 points over the lead baseline, respectively; this demonstrates the effectiveness of using PEER-Document to generate synthetic documents for improving PEER-Edit's ability to explain generated claims by quoting from provided documents.

## 5.4 ITERATIVE EDITING FOR TEXT GENERATION

Finally, we investigate PEER's ability to generate new texts from scratch. To this end, we collect a set of 400 intro sections from Wikipedia, each with three reference documents. As a baseline, we finetune the same language model that was used to initialize PEER as a conditional language model on the exact same data that PEER was trained on – that is, the model is trained to predict $\mathbf{x}_{t+1}$ given $D_t$ and the page's title, but not $\mathbf{x}_t$. However, we use a special character sequence to inform the model about whether $\mathbf{x}_{t+1}$ is an intro section; we train this baseline, that we refer to as *WikiLM*, with the exact same parameters as PEER.

We evaluate PEER in three modes: (i) an *autonomous* mode, where the model continuously writes and realizes its own plans; (ii) a *manual* mode, where we give the model a series of human-written plans. We choose a simple sequence of three plans that we use for all intros: $\mathbf{p}_0$ = "Create page", followed by $\mathbf{p}_1 = \mathbf{p}_2$ = "Add more information"; and (iii) a *collaborative* mode, where human-written plans are interleaved with plans proposed by PEER; that is, we use the plan sequence $\mathbf{p}_0, \mathbf{p}_0', \mathbf{p}_1, \mathbf{p}_1', \mathbf{p}_2$, where $\mathbf{p}_0$, $\mathbf{p}_1$ and $\mathbf{p}_2$ are as above, whereas PEER generates $\mathbf{p}_0'$ and $\mathbf{p}_1'$.

Table 4 shows performance on our test set with length penalties (Murray & Chiang, 2018) optimized on a dev set of 100 intros. While WikiLM performs better than PEER in autonomous mode, PEER in manual mode outperforms WikiLM by about one point Rouge-1, despite using very generic plans that are identical for all intros; the collaborative mode further improves results slightly. To evaluate the faithfulness of all models to the provided documents, we also consider QuestEval (QE) scores (Scialom et al., 2021), which we compute in *reference-less mode*. Interestingly, all variants of PEER perform considerably better in terms of QuestEval scores than WikiLM, suggesting that iteratively updating text helps the model stay more faithful to the provided reference documents.

## 6 ANALYSIS

We take a qualitative look at how PEER performs in a truly *collaborative* setting where we manually provide it with human-written instructions to write both a Wikipedia-style introductory section about itself and a question about guacamole in the *Cooking* forum of StackExchange. For the introductory section, we collect three reference documents containing this paper's author list and affiliation, the abstract, and the caption to Figure 1, respectively. We do not provide any documents for the StackExchange example. Figure 3 shows interactive sessions with PEER (SP, 11B) and PEER (DA, 3B) for writing these texts, respectively. In each step, we generate three different model outputs – one with beam search and two using top-$p$ sampling with $p = 0.9$ – and manually pick one of them.

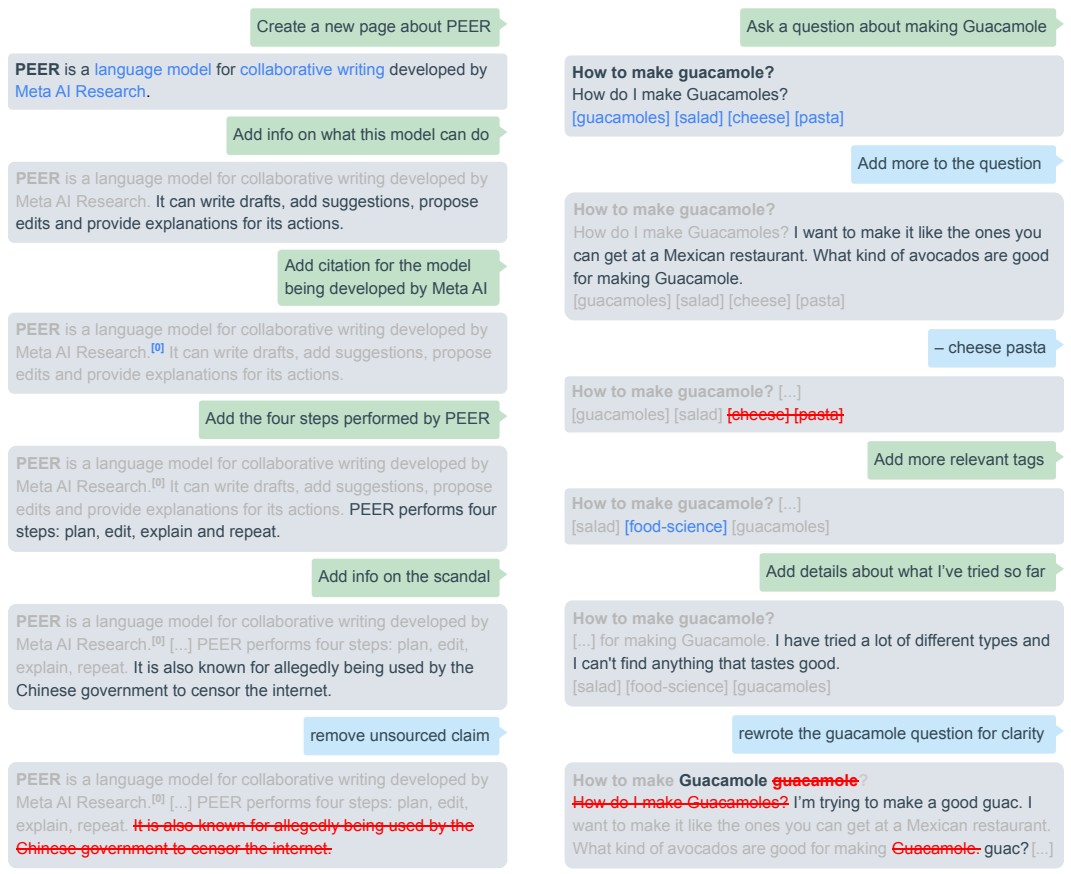

Figure 3: Interactive editing sessions with PEER. Plans on green background are provided by a human, plans on blue background by the model. **Left:** PEER (SP, 11B) writing a Wikipedia-style text about itself. **Right:** PEER (DA, 3B) writing a question in the style of StackExchange.

As can be seen in Figure 3 (left), PEER composes information from various documents to follow provided plans. It makes plausible assumptions, such as the model being developed by Meta AI, despite this not being explicitly stated in any document, and is able to point to the author list (document 0) as a reference. The model's response to the fifth plan ("Add info on the scandal") illustrates a fundamental issue with many LMs: It accepts the premise of this plan and follows it by hallucinating a scandal about internet censorship. However, PEER is able to correct this misinformation in the next step: When not provided with any human-written plan, the model itself writes the plan "remove unsourced claim" and removes the false statement again. Figure 3 (right) shows how after domain adaptation on synthetic edits, PEER is capable of writing and editing texts in domains other than Wikipedia. In particular, it adapts to the structure of questions in StackExchange – consisting of a title (bold), a text, and a sequence of tags – and to their style, which is very different from Wikipedia. PEER proposes plans to fix errors it made in previous steps (such as removing the irrelevant tags "cheese" and "pasta"). It is also able to follow plans like "Add more relevant tags", despite tags being a concept specific to StackExchange that does not occur in its Wikipedia training data.

## 7  CONCLUSION

We have introduced PEER, a language model that can act as a writing assistant by following plans to perform a variety of different textual edits, ranging from syntactic and stylistic edits to changing the meaning of a text by removing, updating or adding information. Through extensive experiments, we have shown that training variants of PEER capable of *infilling* various parts of the editing process enables it to perform edits in different domains, makes it better at following instructions and improves its ability to cite and quote from relevant documents.

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

## A  TRAINING DATA

Our main training data for PEER is derived from Wikipedia's edit history,[5] which directly gives us access to raw tuples of source and target texts $(\mathbf{z}_t, \mathbf{z}_{t+1})$, from which we derive $\mathbf{x}_t$ and $\mathbf{x}_{t+1}$ after some preprocessing steps discussed below. Beyond that, the edit history also provides us with comments $\mathbf{c}_t$ that we use as proxies both for the plan $\mathbf{p}_t$ and for the explanation $\mathbf{e}_t$. Finally, as Wikipedia articles frequently use citations to back up claims, we can obtain an initial set $I_t$ of *document identifiers* (e.g., URLs) for all documents cited in either $\mathbf{z}_t$ or $\mathbf{z}_{t+1}$. Our pipeline for transforming this raw data into the PEER format consists of three steps: First, we use some heuristics for filtering the data to remove low-quality edits and avoid overlap with any of our evaluation sets. We then use $I_t$ and a retrieval engine (Petroni et al., 2022) to obtain a collection of relevant documents $D_t$ for each edit; finally, we convert the data into a format suitable for sequence-to-sequence models. In the following, we discuss all three preprocessing steps in more detail.

---

[5]We use the February 2022 dump available at `https://dumps.wikimedia.org/enwiki/`.

## A.1    FILTERING

As Wikipedia's edit history contains several low quality edits and numerous instances of vandalism (Potthast et al., 2008), we use some simple heuristics to improve data quality. In particular, we filter out edits that were reverted at some point and edits that were automatically made by bots without human involvement. Beyond that, we filter edits that affect more than two paragraphs and remove all edits for pages that are used in any of the datasets we evaluate on. We also discard examples where $I_t$ contains document identifiers that we are unable to resolve (e.g., because they refer to web pages that no longer exist). We filter out revisions with more than 50,000 characters. This makes preprocessing more efficient, as our algorithm for computing diffs between different revisions has squared complexity in the number of characters. Beyond that, we also filter out revision whose comments contain any of the sequences "#", "{{", "}}", "[[", "]]", "template", "image", "infobox" and "pic", as these are usually automatically generated or update parts of the page (such as images and infoboxes) that we remove during preprocessing. We further remove all redirects. Within each chunk of the Wikipedia dump, we downsample revisions for which the corresponding comment occurs in more than 10 revisions so that on average, each comment occurs at most 10 times per chunk. Finally, we filter out edits where either the source paragraphs or the target paragraphs have more than 384 tokens.

## A.2    RETRIEVING DOCUMENTS

A crucial aspect of PEER is its use of documents that contain relevant background information. We thus aim to collect a set of documents $D_t = \{d_t^1, \ldots, d_t^k\}$ for each edit, with $k$ being a hyperparameter defining the number of documents; we set $k = 3$.

To obtain this set of documents $D_t$ for an edit that maps $\mathbf{x}_t$ to $\mathbf{x}_{t+1}$, we make use of the set $I_t$ of document identifiers occuring in $\mathbf{x}_t$ or $\mathbf{x}_{t+1}$. For each document identifier, we get the corresponding document from CCNet (Wenzek et al., 2020). We split the document into non-overlapping chunks of 100 words and use the reranker of Side (Petroni et al., 2022) to find the best chunk given $\mathbf{x}_{t+1}$.

If the number of documents obtained from $I_t$ is below the maximum number of documents per edit, we also use the entire pipeline of Petroni et al. (2022) to find relevant documents in the Sphere corpus Piktus et al. (2021) given $\mathbf{x}_{t+1}$. As this pipeline expects a special `[CIT]` token at the position for which relevant documents are to be retrieved, we place this token right after the first position at which $\mathbf{x}_t$ and $\mathbf{x}_{t+1}$ differ, starting from the right. Note that obtaining documents with this approach requires access to $\mathbf{x}_{t+1}$, so it would be impossible to apply this exact same procedure in real-world settings. However, our focus is not on retrieving relevant documents, but on teaching PEER to perform edits given this information.

## A.3    FORMATTING

Our first formatting step is to remove all paragraphs from $\mathbf{z}_t$ and $\mathbf{z}_{t+1}$ that are not affected by the edit. We then remove Wikipedia-specific syntax, but with a few exceptions: We keep the syntax for representing titles, bold text, text in italics and lists, enabling the model to learn how to perform some basic formatting. We also keep links and, more importantly, citations, enabling PEER to learn how to cite and quote from documents in $D_t$ to back up the textual content it generates. We denote the resulting text sequences with $\mathbf{z}_t'$ and $\mathbf{z}_{t+1}'$.

We linearize each document $d_t^i \in D_t$ using its content $\mathbf{c}_i$ and, if present, its title $\mathbf{t}_i$ and the corresponding web site's domain $\mathbf{d}_i$ as the sequence `[`$i$`]` $\mathbf{d}_i$ `#` $\mathbf{t}_i$ `#` $\mathbf{c}_i$. We include the number $i$ in this representation to facilitate citing and quoting specific documents. To finally obtain $\mathbf{x}_t$ and $\mathbf{x}_{t+1}$ from $\mathbf{z}_t'$ and $\mathbf{z}_{t+1}'$, we replace each citation of a document $d_t^i$ in both sequences with either `[[[`$i$`]]]` or `[[[`$i$ `quote=`$\mathbf{q}_i$`]]]` depending on whether in the original data, a specific subsequence $\mathbf{q}_i$ of $d_t^i$ was quoted. As $\mathbf{p}_t$ and $\mathbf{e}_t$ are already simple text sequences, we do not perform any modifications to them.

If there are multiple inputs or outputs, we simply concatenate them using a special separator sequence. Moreover, if the text we are editing has a title, we always prepend this title to the original input sequence.

In addition to these formatting rules, we randomly remove the page's title for 10% of all examples to make sure that PEER can also work with inputs for which no title is available. We *minimize* 10% of all examples by removing all sentences from both $\mathbf{x}_t$ and $\mathbf{x}_{t+1}$ that are not edited, so that the model also learns to handle and edit single-sentence inputs without context. Finally, to make sure that the model can handle different numbers of reference documents, for 30% of examples we remove $j$ documents from $D_t$, where $j$ is uniformly sampled from $\{1, \ldots, |D_t|\}$. However, we only remove documents that are not cited in either $\mathbf{x}_t$ or $\mathbf{x}_{t+1}$. When linearizing the input and output sequences, for each document $d_t^i \in D_t$, we reserve up to 16 tokens for its domain, 32 tokens for its title, and 196 tokens for the actual content. We truncate all tokens that exceed these limits.

## B   CONTROL TOKENS

To improve the quality and diversity of synthetically generated plans, edits and documents, we implement control mechanisms similar to Keskar et al. (2019) and He et al. (2020b) – that is, we prepend specific *control tokens* to the output sequences that a model is trained to generate, and then use these control tokens during inference to guide the model's generations. In particular, we make use of the following controls for different PEER models:

- For PEER-Explain, we control the output *length* as a proxy for the level of detail in generated explanations. We also control whether the generated comment starts with a verb in infinitive form; this approximates the notion of an *instruction*, the format we expect humans to commonly use for communicating with PEER. Finally, we control whether there is a *word overlap* between the explanation and the edit; preventing this during inference makes sure that generated plans do not make editing trivial by exactly specifying which words to add, remove or replace.

- For PEER-Undo, we control the difference in the *number of words* between $\mathbf{x}_{t+1}$ and $\mathbf{x}_t$. Through this, we can ensure that the sequence $\mathbf{x}_n, \mathbf{x}_{n-1}, \ldots$ eventually terminates at $\mathbf{x}_m = \varepsilon$ and does not get stuck in an infinite loop.

- For PEER-Document, we control whether the generated document *contains a given substring*. This is useful when we want the document to contain a specific quote that is referred to in a Wikipedia edit.

We do not use any controls for PEER-Edit, because – unlike for other models, which have specific and clearly defined tasks to solve – we do not make assumptions in advance about the types of editing tasks that users might want to solve with PEER-Edit and the kinds of control tokens that might be useful for these tasks.

Unlike Keskar et al. (2019), we do not introduce special control *tokens*, but simply express all controls in the form `key=value` where both `key` and `value` are tokenized using the language model's regular tokenizer. In particular, we use the following keys and values:

- `type`: We use this key to control the type of output that PEER-Explain is supposed to generate, with possible values being `instruction` (in which case the output starts with a verb in infitive form) and `other`.

- `length`: This key controls the length of PEER-Explain's output. Values include `s` (less than 2 words), `m` (2–3 words), `l` (4–5 words) and `xl` ($\geq$ 6 words).

- `overlap`: With this key, we control whether there is a word overlap between the edit and the generated output of PEER-Explain; values are `true` and `false`.

- `words`: For PEER-Undo, this key is used to control for the difference in the number of words in $\mathbf{x}_{t+1}$ and $\mathbf{x}_t$; accordingly, the possible values are all integers.

- `contains`: This control can be used to ensure that outputs generated by PEER-Document contain a certain substring, which is provided as the value to this key.

## C   GENERATING SYNTHETIC DATA

For generating synthetic edits, we found it sufficient to apply PEER-Undo just once for each plain text $\mathbf{x}_t$ to obtain a tuple $(\mathbf{p}_{t-1}, \mathbf{x}_{t-1})$. Upon manual inspection, we also found that the generated

plans $\mathbf{p}_{t-1}$ do not actually match the undone edit, so we use PEER-Explain as described in Section 4 to rewrite all plans.

When generating synthetic plans, we use the control tokens discussed in Appendix B to ensure a diverse set of plan lengths. For 80% of generated plans, we enforce that they start with a verb and have no word overlap with the performed edit, respectively.

For obtaining synthetic edits, we sample a single pair $(\mathbf{p}_t, \mathbf{x}_t)$ for each $\mathbf{x}_{t+1}$ using top-$p$ sampling with $p = 0.9$. We sample the value for the `words` control token from a normal distribution with $\mu = -10$ and $\sigma = 8$, clipped at $-40$ and $10$. These values were chosen to allow for a wide range of different values, while also making sure that on average, forward edits *increase* the number of tokens. We rewrite each $\mathbf{p}_t$ with PEER-Explain using the exact same procedure that we use for generating synthetic plans.

For obtaining synthetic plans, we generate 10 different plans with PEER-Explain using top-$p$ sampling with $p = 0.9$. For each pair of $\mathbf{x}_t$ and $\mathbf{x}_{t+1}$, we use a single control sequence for sampling all 10 plans. We choose the `length` uniformly from $\{s,m,l,xl\}$, set `type=instruction` 80% of the time and `overlap=false` 80% of the time.

For obtaining synthetic documents, we sample 10 documents from PEER-Document using top-$p$ sampling with $p = 0.9$, where `contains` is set to the quote from this document that is cited in $\mathbf{x}_{t+1}$. We discard all documents that do not actually contain the quote, and then pick the document that maximizes the probability assigned to the actual edit by PEER-Edit.

## D    TRAINING DETAILS

For training PEER, we use DeepSpeed (Rasley et al., 2020) to enable more efficient multi-GPU training. We use a maximum learning rate of $10^{-4}$, warmup for 2,000 steps and linear decay. We further use gradient clipping with a maximum norm of 1.0, weight decay of 0.01 and a dropout rate of 0.1. The maximum sequence length is set to 1,024 and 384 tokens for input and output, respectively.

## E    EVALUATION DETAILS

We use a variety of metrics to evaluate PEER and our baseline models on all tasks considered:

- **Exact Match** (EM) is the percentage of examples for which the performed edit exactly matches a given target;
- **EM-Diff** is a variant of EM that is computed on the diff level;[6]
- **SARI** (Xu et al., 2016) averages match scores for the three word-level edit operations *add*, *delete* and *keep*;[7]
- **GLEU** (Napoles et al., 2015) is a variant of BLEU (Papineni et al., 2002) proposed for grammatical error correction tasks;
- **Rouge** (Lin, 2004) is a set of metrics based on $n$-gram overlap (Rouge-$n$) or longest common subsequences (Rouge-L);
- **Update-Rouge** (Logan IV et al., 2021) is a variant of Rouge that is computed only on sentences updated during an edit.

### E.1    NATURAL EDITS

In addition to the experiments conducted in Section 5.1, we check whether PEER actually learns to make use of provided documents and plans by evaluating it on the Wikipedia subset of Natural Edits. We compare regular PEER provided with gold plans to variants trained and evaluated (i)

---

[6]Diffs are obtained using Python's `difflib` library. For a model output $\mathbf{x}'_{t+1}$, we compute EM-Diff as $|\text{diff}(\mathbf{x}_t, \mathbf{x}_{t+1}) \cap \text{diff}(\mathbf{x}_t, \mathbf{x}'_{t+1})| \div \max(|\text{diff}(\mathbf{x}_t, \mathbf{x}_{t+1})|, |\text{diff}(\mathbf{x}_t, \mathbf{x}'_{t+1})|)$.

[7]We use the SARI implementation of EASSE (Alva-Manchego et al., 2019).

Table 5: Results for variants of PEER on the Wikipedia subset of Natural Edits. Plans and documents provide complementary information and substantially improve performance.

| Model | EM | EM-Diff | SARI |
|---|---|---|---|
| Copy | 0.4 | 0.0 | 32.7 |
| PEER | **23.1** | **26.2** | **55.5** |
| PEER (no plans) | 18.0 | 19.8 | 52.0 |
| PEER (no documents) | 19.8 | 22.8 | 51.7 |
| PEER (no plans/documents) | 13.5 | 15.1 | 45.9 |

Table 6: Overview of the number of edits and plain texts (PT) in the train sets and the number of edits in the test sets of Natural Edits. The final column shows whether the subset uses reference documents.

| Subset | Train (Edit) | Train (PT) | Test | Doc. |
|---|---|---|---|---|
| Wikipedia | 6,960,935 | – | 4,000 | ✓ |
| Wikinews | – | 125,664 | 1,000 | – |
| Cooking | – | 22,517 | 500 | – |
| Gardening | – | 13,258 | 500 | – |
| Law | – | 16,418 | 500 | – |
| Movies | – | 19,601 | 500 | – |
| Politics | – | 10,676 | 500 | – |
| Travel | – | 38,961 | 500 | – |
| Workplace | – | 18,231 | 500 | – |

without plans, (ii) without reference documents, and (iii) without both plans and reference documents. Table 5 shows EM, EM-Diff and SARI scores for all models and a copying baseline, for which $\mathbf{x}_{t+1} = \mathbf{x}_t$. As can be seen, PEER substantially outperforms all baselines. PEER without both plans and documents performs much worse than just removing one of both, illustrating that plans and documents provide complementary information that the model is capable of using; this is in line with findings of Faltings et al. (2021).

## E.2 DOWNSTREAM TASKS

The plans used for each of the downstream tasks considered in Section 5.2 are shown in Table 7. We manually wrote instructions for all datasets except ITERATER, for which we directly took instructions from the definitions provided by Du et al. (2022b).

For most baseline models (T0, GPT3, InstructGPT and OPT), we wrap each plan $\mathbf{p}$ for an input $\mathbf{x}_t$ with the following template:

```
Task: p
Input: x_t
Output:
```

Table 7: Plans used for the downstream tasks considered in Section 5.2

| Task | Plan |
|---|---|
| JFLEG | Fix grammar errors |
| ASSET | Simplify this sentence |
| ITERATER (fluency) | Fix grammatical errors in the text. |
| ITERATER (coherence) | Make the text more cohesive, logically linked and consistent as a whole. |
| ITERATER (clarity) | Make the text more formal, concise, readable and understandable. |
| WNC | Remove POV |
| FRUIT | Update the article |
| WAFER-INS | Add missing information |

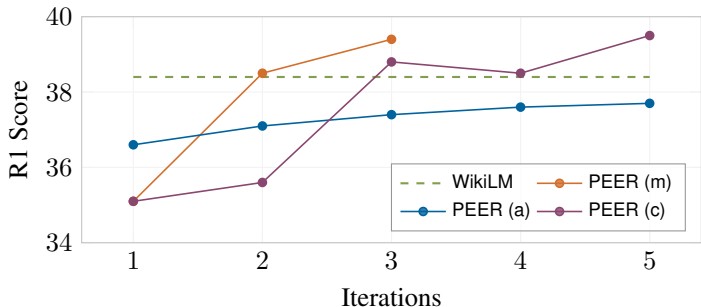

Figure 4: Average Rouge-1 score of WikiLM and PEER in autonomous (a), manual (m) and collaborative (c) mode as a function of the number of iterations

For T$k$-Instruct, we replace the string "Task" with "Definition" to match their format. For tasks that require references, we additionally add all references following the string "Reference:" after the input. For examples that also provide a title, we add this title following the string "Title:" before the input.

### E.3 CITING AND QUOTING

Naturally, we make sure that the cited document is always present in the $k = 3$ documents provided to PEER. To make the task of citing the correct document challenging, we obtain the other two documents in $D_t$ by applying the BM25 (Robertson et al., 1995) and DPR (Karpukhin et al., 2020) variants of Petroni et al. (2022) to find the best match in Sphere (Piktus et al., 2021), respectively. If the gold document contains too many tokens, for NE-Cite we pick the best chunk according to the reranker of Petroni et al. (2022); for NE-Quote, we select the chunk from the document that actually contains the quote. In total, we collect 2,351 and 391 examples, respectively, for which we manually set the plans to simply be "Add a citation" and "Add a quote".

### E.4 ITERATIVE EDITING FOR TEXT GENERATION

Without controlling for output length, WikiLM generates rather short intros, resulting in relatively low Rouge-1/2/L scores. To make the comparison more fair, we thus split our dataset of Wikipedia intros into 100 dev examples and 400 test examples; the dev examples are exclusively used for picking the exponential *length penalty* (Murray & Chiang, 2018) that maximizes the model's average Rouge-1 score. We also prevent models from generating the same token 5-gram more than once to avoid endless repetitions.

Figure 4 shows how performance for different PEER modes changes across iterations, illustrating how generated intros are improved over multiple iterations.

## F ANALYSIS

For the introductory section, we collect three reference documents $d_0$, $d_1$, and $d_2$, where the first document contains this paper's author list and affiliation, the second document contains the abstract, and the third document contains the caption to Figure 1. For all documents, we set the title to this paper's title and the domain to `arxiv.org`. We use this same set of documents for each generation step, i.e., $D_t = \{d_0, d_1, d_2\}$ for all $t$. We do not provide any documents for the StackExchange example. Figure 3 shows interactive sessions with PEER (SP, 11B) and PEER (DA, 3B) for writing these texts, respectively. In each step, we generate three different model outputs – one with beam search using three beams, and two using top-$p$ sampling with $p = 0.9$ – and manually pick one of them.

To better understand the quality of the synthetic data generated with our infilling procedure, we also look at exemplary outputs of the other PEER variants. We first consider PEER-Undo, the model we use to generate edits for domains where only plain texts are available. Figure 5 shows

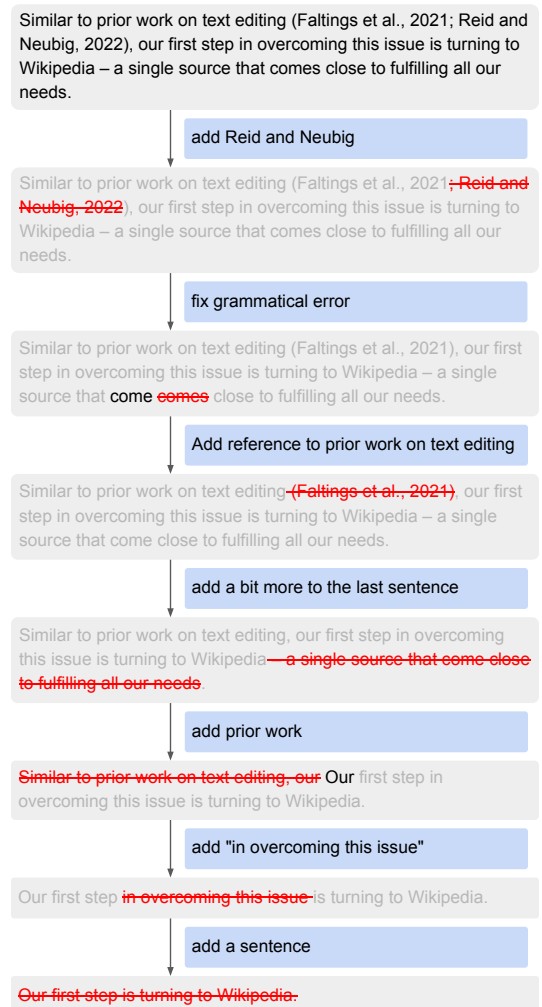

Figure 5: Exemplary application of PEER-Undo for decomposing a sentence from this paper into multiple edits, terminating with an empty sequence. Plans are rewritten with PEER-Explain in the opposite direction.

the result of iteratively applying PEER-Undo to a selected sentence from this paper; corresponding plans are obtained from PEER-Explain. As can be seen, PEER-Undo is able to decompose this sentence into a sequence of meaningful edits despite not being exposed to any scientific papers during training. Somewhat surprisingly, both PEER-Undo and PEER-Explain are able to handle the references contained in this sentence, despite them being formatted in a completely different way than how we represent references during training on Wikipedia data (i.e., replacing them with a numeric identifier in square brackets).

We next inspect PEER-Explain's ability to generate plans as discussed in Section 4. For an edit that we performed while writing this paper, Table 8 shows the plans generated with PEER-Explain for different control sequences, using greedy decoding (see Appendix B for details on control sequences). As can be seen, length is a reasonable proxy for the amount of details in a plan: Constraining the output to be short results in the plan "add citation", whereas for a greater output length, PEER-Explain correctly identifies that *two* changes were made ("add citation for JFLEG and add a bit more detail"). Allowing word overlap between the plan and the edit results in a plan that specifies exactly which reference to add ("add reference to Napoles et al., 2017"). The final column of Table 8 shows the average probability of tokens in $\mathbf{x}_{t+1}$ according to PEER-Edit given both $\mathbf{x}_t$ and

Table 8: Generated plans for an edit that we performed while writing this paper; the corresponding sequences $\mathbf{x}_t$ and $\mathbf{x}_{t+1}$ are shown on top, with changes highlighted in green. The final columns shows the average probability across tokens that our main PEER model assigns to $\mathbf{x}_{t+1}$ given $\mathbf{x}_t$ and the respective plan. Control sequences enable us to specify the level of detail for a plan.

$\mathbf{x}_t$ = JFLEG is a grammatical error correction dataset with single-sentence inputs.
$\mathbf{x}_{t+1}$ = JFLEG (Napoles et al., 2017) is [. . .] inputs written by English language learners.

| Control Sequence | Output | Score |
|---|---|---|
| `type=instruction length=s`
`overlap=false` | add citation | 0.16 |
| `type=instruction length=m`
`overlap=false` | add reference to JFLEG | 0.15 |
| `type=instruction length=xl`
`overlap=false` | add citation for JFLEG and add a bit more detail | 0.17 |
| `type=instruction length=xl`
`overlap=true` | add reference to Napoles et al., 2017 | **0.26** |
| `type=other length=xl`
`overlap=false` | Added a reference to the JFLEG paper | 0.16 |

Figure 6: A document generated with PEER-Document for an edit that slightly modifies $\mathbf{x}_t$ (top) by adding a citation to some document with id 0 at the very end. As a control, we enforce that the document contains the substring "outperforms PEER on Natural Edits". The generated reference backs up the claim but contains a lot of repetitions.

$\mathbf{x}_t$ = Importantly, the domain-adapted variants of PEER clearly outperform regular PEER for all subsets of Natural Edits.

---

**[0] Domain-Adapted PEER for Natural Edits — Springer for Research & Development**
`rd.springer.com`
Domain-Adapted PEER for Natural Edits The main goal of this work is to develop a domain-adaptive variant of PEER, which outperforms PEER on Natural Edits with respect to both the number of Natural Edits and the number of Natural Edits with respect to the number of natural edits. In this paper, we present a domain-adaptive

---

each of the generated plans. Naturally, the plan with word overlap is most helpful, resulting in the highest score; all other plans are about equally helpful to PEER-Edit.

We finally look at the ability of PEER-Document to generate plausible-looking reference documents, once again using an example sentence from this paper. For a synthetic edit that just adds a citation at the very end of this sentence, we sample five outputs from PEER-Document; the best generated document among these five is shown in Figure 6. As can be seen, the model is able to produce a somewhat plausible reference document that provides evidence for the claim $\mathbf{x}_t$. However, as exemplified by this document, we found model outputs to often contain numerous repetitions ("the number of Natural Edits and the number of Natural Edits").

# G    HUMAN EVALUATION

## G.1    QUALITY OF GENERATED EXPLANATIONS

In order to further evaluate the ability of PEER-Explain to provide good explanations for edits, we perform a small-scale human evaluation. To this end, we randomly selected 50 Wikipedia edits with both their original comments and an explanation generated by PEER-Explain.[8] Without knowing which explanation is generated by our model and which explanation is derived from the original

---

[8] In accordance with the procedure described in Section 4, we obtained the explanation from PEER-Explain by randomly sampling 10 different explanations and keeping the one that maximizes the probability of PEER-Edit performing the actual edit.

comment, three authors of this paper selected their preferred explanation for each edit, based on *how well it describes the edit being made*. Overall, the original comment was preferred 28% of the time, whereas the explanation generated by PEER-Explain was preferred 59% of the time; in 13% of cases, both explanations were judged as being equally good. Inter-rater agreement was at 71%; this may be due to the fact that annotators were not provided with a precise definition of what constitutes a good explanation.

### G.2 QUALITY OF GENERATED DOCUMENTS

To assess the quality of documents generated by PEER-Document, we randomly sampled 50 edits containing citations with quotes for which our training corpus does not contain the corresponding reference document. We used PEER-Document to generate new document snippets for each of them; three authors of this paper annotated all generated documents by answering the following three questions:

- **Relevance**: Does the document support the claims made in the updated text?
- **Fluency**: Does the document sound like natural text found on an English webpage?
- **Coherence**: Is the document coherent and consistent?

Overall, we found 75% of the generated documents to be relevant, 89% to be fluent and 83% to be coherent. Inter-annotator agreement was consistently around 80% (88%, 84% and 80% for relevance, fluency and coherence, respectively). These results indicate that the majority of generated documents is of reasonably high quality to be used as synthetic data for finetuning PEER-Edit.

### G.3 ERROR ANALYSIS

To get a better understanding of the kinds of errors that PEER-Edit makes when proposing edits, we manually looked at 100 input texts and plans that we randomly sampled from the Wikipedia subset of Natural Edits, and corresponding edits generated by PEER (SP, 11B). For this, we only considered examples where the edit proposed by PEER is *not* identical to the gold edit as we are interested in failure cases of PEER. Of the 100 edits selected, we found 50% to not contain any error. In total, 16% of the generated edits contained hallucinations; however, for 10 out of these 16 examples, the model was actually forced to hallucinate in order to follow the given instruction, because following the instruction required adding new information (such as a date of birth) that was not present in any of the provided reference documents. For another 10% of the considered examples, PEER-Edit did either not change the text at all, or only partially follow the instruction. In 5% of cases, PEER-Edit made changes largely unrelated to the specified instructions and in 7% of cases, the generated edit contained multiple repetitions of the same phrase. Finally, for 12 out of 100 examples, the given instructions were unclear or extremely vague, making it impossible for PEER to follow them.

## H LIMITATIONS

A major limitation of our approach is that at each editing step, we assume the set $D_t$ to be given; the retrieval engine we use to obtain $D_t$ (Petroni et al., 2022) makes use of the targets $\mathbf{x}_{t+1}$, which clearly is not possible in real-world applications. It would thus be interesting to investigate how incorporating a retrieval engine that does not have access to $\mathbf{x}_{t+1}$ or even jointly training it along with the model, as is done by Guu et al. (2020) and Borgeaud et al. (2021), would affect results.

Despite being able to use reference documents and obtaining comparably high QuestEval scores in our intro generation experiments, upon manual inspection we still found PEER to generate false statements or claims not backed up by the provided documents in many cases. While the ability to cite and quote generally makes it easier to check such hallucinations, citations can also make the model's generations appear more authoritative, thus making it more likely that users rely on them without explicit fact checking (Nakano et al., 2021).

Finally, we use a very simple approach for representing edits by rewriting the entire paragraph. This makes PEER less efficient than other recent approaches for editing (Logan IV et al., 2021; Reid & Neubig, 2022); also, our inefficient way of representing both inputs and outputs makes it impossible to handle entire documents, which we believe to be crucial for many real-world applications.

Our evaluation is limited in that it only evaluates PEER and other models on a small subset of potential editing tasks in few different domains; all evaluations are performed in English only. Besides, we also explore the collaborative potential of PEER only in a very limited way: While arguably, the ability to follow human-written plans and perform a variety of edits both with and without reference documents (Table 2) in different domains (Table 1), to cite and quote (Table 3), and to autonomously generate plans (Table 4) are important building blocks of a collaborative model, it would be interesting for follow-up work to consider entire sessions of human-AI interactions beyond individual examples like the one shown in Figure 3. However, this requires solving many of the challenges discussed previously, such as having access to an actual retrieval engine that can obtain relevant documents on the fly, finding suitable ways of evaluating texts jointly authored by humans and language models, and improving PEER's efficiency to enable processing entire documents.

