# OpenReview forum: "PEER: A Collaborative Language Model"
_ICLR.cc/2023/Conference — ICLR 2023 notable top 25%_

### Official Review · Reviewer_WDUk · 2022-10-21

**Confidence:** 4
**Correctness:** 4
**Technical Novelty And Significance:** 3
**Empirical Novelty And Significance:** 4
**Recommendation:** 6

**Clarity, Quality, Novelty And Reproducibility:**

The paper is well-written and centered around the application of collaborative editing. Descriptions were clear in most places; some less clear aspects were on the technical novelty, benefits compared to simple ICL baselines, and generalizability/science impact beyond making the collaborative application possible.

The novel aspects of this work include decomposing problem and training pretrained LMs on objectives for different writing purposes (edit, undo, explain, generate document). Also, leveraging them to augment data for domains that lack editing and document histories. Overal, interesting approach for a proof-of-concept but potential for adoption might be limited due to technical limitations (e.g. training data assumptions, lack of comparison to simple alternatives).

The data and code will be released so it should be possible to reproduce the results in this work; I don't have any major concerns on this front since training details are also included in the supplementary.

**Strength And Weaknesses:**

**Strengths**

Idea of training LMs to perform collaborative writing is interesting and the application exciting. Current tools for editing do not support such advanced functionalities and could have large  commercial potential.

Showcases feasibility of collaborative writing with pretrained LMs (proof of concept) and improvements over unsupervised pretrained LMs and LMs that have been trained on a broad set of human instructions.

**Weaknesses**

The approach requires training different LMs for four functionalities which is not ideal from maintainability and efficiency standpoint. Discussion about this design choice versus pretraining a single model to perform different actions would be useful.

Assumption is that a large dataset of annotation is available for training and access to citations/documents which are not realistic. Even though an approach for extending to other domains is proposed, it still relies on annotated data and additional costly steps for data augmentation and training. This limitation has not been examined or discussed thoroughly in the paper.

Lack of baselines with vanilla in-context learning (ICL) with editing/performing undo/explaining examples. There have been a number of papers demonstrating that ICL can work surprisingly well. It would be interesting to compare with simple in-context learning with a few examples of edits for instance.


**Summary Of The Paper:**

The paper focuses on collaborative writing based on pretrained LMs. In particular, it proposes a collaborative LM that writes drafts, adds suggestions, proposes edits, and provides explanations. To training is performed on edits and cited documents (at the next step) that is assumed to be available a priori as well as an infilling technique that allows generating data for new domains to increase applicability and training diversity. The evaluation focuses on planning in settings with edits and histories available across different domains, downstream editing tasks with human instructions, and quoting & citing, shows improved performance compared to unsupervised baselines or baselines pretrained on human instructions.

**Summary Of The Review:**

An interesting proof-of-concept paper centering around an exciting application: collaborative writing with pretrained LMs. The approach has novel aspects such decomposing problem and synthesizing data for planning but it makes some unrealistic assumptions about the data availability and lacks comparison to simple few-shot baselines. Synthesizing data for planning should be useful for the research community but the technical part overall might have limited impact for the reasons explained above.

---

> ### Author Response · Authors · 2022-11-17
> **Need for Annotated Data**
>
> > The approach requires training different LMs for four functionalities which is not ideal from maintainability and efficiency standpoint. Discussion about this design choice versus pretraining a single model to perform different actions would be useful.
>
> In preliminary experiments, we also tried a variant where we train a single LM to perform all four functionalities; however, we found this to perform worse than having four separate models; we have added this information to the paper. While we agree that having four different models is not ideal in terms of maintainability and efficiency, note that PEER-Document and PEER-Undo are only necessary during training and can be discarded afterwards.
>
> > Assumption is that a large dataset of annotation is available for training and access to citations/documents which are not realistic. Even though an approach for extending to other domains is proposed, it still relies on annotated data and additional costly steps for data augmentation and training. This limitation has not been examined or discussed thoroughly in the paper.
>
> Importantly, our approach for transferring PEER to other domains does *not* require access to any annotated data – only the training of our original PEER model on Wikipedia requires annotated data. As shown in Table 2, we adapt PEER to various StackExchange forums using only plain texts without any edit histories, documents or plans. All the missing information is synthetically generated using the different PEER variants. While our approach of transferring PEER to other domains is costly (due to the data augmentation steps), that cost occurs only once.
> > Lack of baselines with vanilla in-context learning (ICL) with editing/performing undo/explaining examples. There have been a number of papers demonstrating that ICL can work surprisingly well. It would be interesting to compare with simple in-context learning with a few examples of edits for instance.
>
> In the general editing setup that we consider for PEER (where the input consists of a paragraph to be edited and multiple reference documents), a single input requires hundreds of tokens in most cases. As T5 only supports a maximum sequence length of 1024 tokens, it is not possible to fit additional in-context examples, which would make the sequence even longer (in addition to the input and reference documents, the in-context examples would also have an output).

---

### Official Review · Reviewer_fvQS · 2022-10-22

**Confidence:** 4
**Correctness:** 3
**Technical Novelty And Significance:** 3
**Empirical Novelty And Significance:** 3
**Recommendation:** 8

**Clarity, Quality, Novelty And Reproducibility:**


- There is a recently released and co-occurrent paper that is very relevant to this work: EDITEVAL: An Instruction-Based Benchmark for Text Improvements. People can take it as a reference.
- Even though some baselines (e.g., Text editing by command) only do sentence-level editions, their model can still give better numbers than “copy”. You can either repeatedly put commands to each sentence or just treat a few sentences together as one single long string.
- Better to provide evaluation metrics to the main paper, e.g., SARI is not that familiar to me, at least add a citation there.
- The novelty of the modeling side is minor, and the authors do not explain clearly what their strategy is to determine data augmentation for weakly-supervised setting in the main paper. (Actually, I just found there is one section in Appendix C, which can answer some of my questions. )

**Strength And Weaknesses:**

S:
- The idea of controlling the generation of language models step-by-step in a recurred manner is interesting.
- The authors have evaluated their models on multiple settings, proving that their models trained on Wikipedia can potentially generalize to multiple downstream tasks.

W:
- Human evaluation is missing and could add more insights to the interactive process.
- Even though there is a “explain” component in PEER, it is not evaluated and studied regarding its correctness.
- No error analysis about the generated plans and the edited text. The authors only put a sentence at the end of the Appendix saying that “we still found PEER to generate false statements or claims not backed up by the provided documents in many cases“, but in the main paper there is no discussion or statistics on this weakness.

**Summary Of The Paper:**

This paper introduces a collaborative language model setup trained on Wikipedia edition history. PEER stands for plan, edit, explain, and repeat. The work is closely related to text editing and instruction LM literature. The paper makes an assumption that the grounded knowledge for editing is given. The authors trained several variances models with T5 including PEER-Edit, PEER-undo, PEER-explain, and PEER-Document, leveraging some of them to augment training data. They evaluated their models on several setting such as Natural Edits with comments, 0-shot testing on downstream tasks, citation and quote, and iterative editing. They showed their models are better than baseline with similar size and even better than some LLMs.

**Summary Of The Review:**

Overall, I think this is an exciting paper that tried to improve the usability of controllable language modeling. Making LMs controllable and interactive is definitely an essential next step for our field to move forward. My biggest concern is evaluation. It is hard to persuade people this model is ready without solid human evaluation since this task could be open-ended: the edition is the actual usage can be ambiguous, and the same plan can lead to multiple different outputs, similar to open-domain dialogues.

---

> ### Author Response · Authors · 2022-11-17
> **Human Evaluation**
>
> > Human evaluation is missing and could add more insights to the interactive process.
>
> We agree that human evaluation of the interactive process would be very interesting for follow-up work. We discuss the limitations of our evaluation and our reasons for not conducting human evaluation in the last paragraph of Appendix H.
>
> > Even though there is a “explain” component in PEER, it is not evaluated and studied regarding its correctness.
>
> In the updated version of our paper, we have added a small-scale human evaluation, where we manually assess both documents and plans/explanations generated by different PEER models (Appendix G). The main insight of our human evaluation regarding the "explain" component is that humans prefer explanations written by PEER-Explain over the original comments found in Wikipedia in 59% of cases. Additionally, the utility of PEER-Explain is evaluated indirectly as we train PEER-Edit on plans generated by PEER-Explain; Table 2 shows clear improvements for the variant of PEER finetuned on synthetic plans produced by PEER-Explain (“PEER” vs “PEER (SP)”), so apart from providing explanations, the “explain” component is also useful as it improves the editing model’s ability to follow instructions.
>
> > No error analysis about the generated plans and the edited text. The authors only put a sentence at the end of the Appendix saying that “we still found PEER to generate false statements or claims not backed up by the provided documents in many cases”, but in the main paper there is no discussion or statistics on this weakness.
>
> We have added a small-scale error analysis where we manually looked at some outputs of PEER and categorized errors. Typical errors include hallucinations (16%) – especially when the plan forces PEER to add information that is not present in any of the provided reference documents –, not changing the input text at all or only partially following an instruction (10%), and introducing repetitions (7%). More details can be found in Appendix G.3.

---

### Official Review · Reviewer_tmW2 · 2022-10-24

**Confidence:** 5
**Correctness:** 4
**Technical Novelty And Significance:** 4
**Empirical Novelty And Significance:** 4
**Recommendation:** 8

**Clarity, Quality, Novelty And Reproducibility:**

Clarity&Quality: The paper is well-written and easy to follow.
Novelty: Good novelty -- new setting & model (collaborative editing) are developed. A new dataset is introduced.
Reproducibility: The authors didn't provide the code so we cannot reproduce it.


**Strength And Weaknesses:**

Strength:
- This paper was the first to combine multiple collaborative writing skills together into a language model.
- Solid experiment results show its effectiveness on various editing tasks in a zero-shot fashion and also citing and quoting tasks.
- Enable collaborative editing for generating text as shown in Table 4.
- New datasets: Natural Edits, NE-Cite and NE-Quote are introduced.

Weaknesses:
- Many of your downstream tasks include Wikipedia, which may be included in the training set of T5. Is there any way to prevent this issue?

**Summary Of The Paper:**

The authors proposed PEER, a language model that includes four skills: plan, edit, explain, and repeat. To better use the Wikipedia edit history with missing parts, the author proposed four infilling operations to overcome this issue: PEER-edit, PEER-undo, PEER-explain, PEER-document. The authors utilize the pretrained T5 to initialize PEER and train on Wikipedia edits. The experiment result shows the effectiveness of PEER on the Natural Edits and many downstream datasets.

**Summary Of The Review:**

In general it's a good paper with solid results. I would like to recommend this paper be accepted.

---

> ### Author Response · Authors · 2022-11-17
> **Overlap with Wikipedia**
>
> > Many of your downstream tasks include Wikipedia, which may be included in the training set of T5. Is there any way to prevent this issue?
>
> We think that this issue is hard to circumvent, given that most pretrained language models have Wikipedia in their training sets. However, T5 is only trained on a single snapshot of Wikipedia (as opposed to the entire edit history). Also, all of our baselines are either also T5-based or their training data contains Wikipedia, so we believe that this does not give an unfair advantage to PEER.
>
> > The authors didn’t provide the code so we cannot reproduce it.
>
> We’re currently working on making the code for training and inference as well as all scripts for pre-processing publicly available. We also discuss details of pre-processing, training and evaluation in Appendix A, D and E, which hopefully helps with reproducing our results.

---

### Official Review · Reviewer_eo6j · 2022-10-26

**Confidence:** 4
**Correctness:** 4
**Technical Novelty And Significance:** 3
**Empirical Novelty And Significance:** 4
**Recommendation:** 8

**Clarity, Quality, Novelty And Reproducibility:**

This paper is overall a high-quality paper with clear clarification and novelty mainly on the applicational side. The reproducibility is degraded due to the complex pre-processing steps of datasets and the lack of codes.

**Strength And Weaknesses:**

Strengths:

1) Collaborative language modeling is an interesting and promising direction in NLG, which increases the interpretability and controllability of traditional NLG models. This direction also connects with the industrial products about writing assistants, making the technique of NLG more applicable.
2) The proposed method based on self-training and instruction tuning is simple and effective, which directly supports the motivation to imitate the collaborative writing process.
3) The experiment part is well organized. The four research questions are essential for collaborative language modeling. And the authors provide empirical results on diverse tasks and datasets to successfully answer each question.

Weaknesses:

1) Since PEER contains four different models which infill the missing part of training data, the authors should test the performance of each model (including PEER-Edit, PEER-Undo, PEER-Explain, and PEER-Document) to demonstrate the quality of augmented data. Especially the performance of PEER-Document should be shown and analyzed, because in my view it’s hard for a pre-trained model with 3B model parameters to generate high-quality knowledge documents only given the texts before / after editing and plans.

2) More baselines should be included in the main experiments of text editing. For example, Table 1 only contains a copy-based baseline, which may exaggerate the improvement of PEER. At least the Wikipedia subset which contains the edit histories should have more baselines.

3) In this paper, the plan is just defined as a short text sequence like instructions. This setting seems a little bit toy because it doesn’t clearly describe which part should be edited. This may put more burden on the design of instructions. From Figure 3, I find that the instructions mostly contain the position information, such as “add citation for the model being developed by Meta AI”. But it’s obviously hard when the text is long and has many objectives to describe. Thus, the design of plans should be further discussed to make this scenario more realistic.

4) Typo: the second x_t –> x_{t+1} on the right side of Figure 2 (PEER-Document)

**Summary Of The Paper:**

This paper proposes a collaborative language model called PEER which is trained to imitate the process of collaborative writing. PEER with four T5 models can support the actions of editing, undo, adding explanation, and document generation. To overcome the problem of data scarcity and improve the generalization ability of PEER, the authors adopt self-training to infill the intermediate parts of training data, which increases the quality, amount and diversity of training data. Experimental results show the effectiveness of PEER across various domains and editing tasks.

**Summary Of The Review:**

This paper proposes a novel collaborative language model and gives strong empirical results. I think it’s an insightful paper for NLP community and will recommend acceptance if the authors can successfully solve my concerns in the rebuttal.

---

> ### Author Response · Authors · 2022-11-17
> **Human Evaluation and Clarifications**
>
> > The authors should test the performance of each model to demonstrate the quality of augmented data. Especially the performance of PEER-Document should be shown and analyzed.
>
> We agree that it is challenging for 3B parameter models to generate high-quality documents. However, the sole purpose of the documents is to improve the ability of PEER-Edit to cite and quote from documents; numbers in Table 3 show that the quality of the documents is high enough to fulfill this purpose. To provide further insights into the quality of the augmented data, we have added a small-scale human evaluation, where three authors of the paper manually assessed both documents and explanations generated by different PEER models (Appendix G). The main insights of this human evaluation are that (i) most of the documents generated by PEER-Document are relevant (75%), fluent (89%) and coherent (83%), and (ii) humans prefer explanations written by PEER-Explain over the original comments found in Wikipedia in 59% of cases. We have also added a human evaluation of errors made by PEER-Edit (Appendix G.3).
>
>
> Note that (due to T5’s maximum context length of 1024 tokens) we truncate each document after at most 196 tokens (cf Appendix A), and writing relevant and coherent document snippets of 196 tokens is, of course, an easier task than writing full documents, so even a model with 3B parameters can do this reasonably well.
>
> > Table 1 only contains a copy-based baseline, which may exaggerate the improvement of PEER.
>
> We would like to clarify that the main purpose of Table 1 is to quantify the effectiveness of our domain-adaptation approach by comparing PEER and domain-adapted PEER. Because Natural Edits is extremely domain specific, domain-adapted PEER in particular would likely have a clear advantage over other baseline models not trained on data of the same domain. Consequently, we instead refer to Table 2, which presents a comprehensive evaluation of PEER against other models on downstream tasks that are much less domain-specific. Table 5 (Appendix E) contains some more baselines (variants of PEER without plans and/or documents) which quantify the importance of using plans and documents.
>
> > The plan is just defined as a short text sequence like instructions. This setting seems a little bit toy because it doesn’t clearly describe which part should be edited. [...] It’s obviously hard when the text is long and has many objectives to describe. Thus, the design of plans should be further discussed to make this scenario more realistic.
>
> We exclusively focus on paragraph-level edits, meaning that the text will not exceed a certain length (but we believe looking at full documents is an exciting problem for future work). Within the paragraph, it is the model’s responsibility to find the best part to do an edit (unless explicitly specified by the user): For example, if the model is told to fix grammar errors, it needs to identify the sentences in the paragraph that contain such errors; if it is told to insert information, it needs to pick the best place to insert this information itself.
>
> > The reproducibility is degraded due to the complex pre-processing steps of datasets and the lack of codes.
>
> We’re currently working on making the code for training and inference as well as all scripts for pre-processing publicly available. The pre-processing steps are also discussed in detail in Appendix A.

---

### Decision · Program_Chairs · 2023-01-20

**Decision:**

Accept: notable-top-25%

**Justification For Why Not Higher Score:**

While this is a great paper as agreed by the reviewers, the application domain seems a little too niche to justify an oral.

**Justification For Why Not Lower Score:**

All of the reviewers agree that this is a great paper, and it produces good results on a task that could be useful in practice to real users, so it justifies being highlighted with a talk.

**Metareview: Summary, Strengths And Weaknesses:**

This paper proposes a method for enabling a language model to edit text. Text editing is split into 4 tasks that can be iteratively performed until no further edits are required. These are Edit, Undo, Explain, and Document. Edit takes in a paragraph of text and relevant documents and produces an edit plan and an updated version of the text. Undo is the inverse of Edit, it produces a plan and a previous version of the paragraph given the updated paragraph and the documents. Explain produces an explanation (similar to the plan) given the two versions of the paragraph and the documents. Document produces the documents given the two versions of the text and the plan. These models can not only be used to do edits at inference time, but also to generate the training data to generalise PEER to other domains than Wikipedia (the original data that the component parts of PEER were trained on).

The authors propose a general method for an important application - text editing, which can be of interest to real users.

One weakness of the paper is that it requires training 4 different models to work at its best, which can be brittle. The paper is also missing prompted baselines. While the authors included a small human evaluation study to measure the quality of the edits produced by PEER, a more thorough analysis would be good to have.


**Note From Pc:**

if the above contains the word "oral" or "spotlight" please see: "oral" presentation means -> notable-top-5% and "spotlight" means -> notable-top-25%. As stated in our emails, we are disassociating presentation type from AC recommendations